

# Polycaprolactone nanofibers as an adjuvant strategy for Tamoxifen release and their cytotoxicity on breast cancer cells

Ana D. Pinzón-García[1], Ruben Sinisterra[1], Maria Cortes[2], Fredy Mesa[3] and Sandra Ramírez-Clavijo[3]

[1] Chemistry Department, Instituto de Ciências Exatas, Universidade Federal de Minas Gerais, Belo Horizonte, Brazil
[2] Restorative Dentistry Department, Faculty of Dentistry, Universidade Federal de Minas Gerais, Belo Horizonte, Brazil
[3] Faculty of Natural Sciences, Department of Biology, Universidade del Rosario, Bogotá, Colombia

Corresponding author
Fredy Mesa,
fredy.mesa@urosario.edu.co

## ABSTRACT

Breast cancer is the second leading cause of death in women, and tamoxifen citrate (TMX) is accepted widely for the treatment of hormone receptor–positive breast cancers. Several local drug-delivery systems, including nanofibers, have been developed for antitumor treatment. Nanofibers are biomaterials that mimic the natural extracellular matrix, and they have been used as controlled release devices because they enable highly efficient drug loading. The purpose of the present study was to develop polycaprolactone (PCL) nanofibers incorporating TMX for use in the treatment of breast tumors. Pristine PCL and PCL-TMX nanofibers were produced by electrospinning and characterized physiochemically using different techniques. In addition, an in vitro study of TMX release from the nanofibers was performed. The PCL-TMX nanofibers showed sustained TMX release up to 14 h, releasing 100% of the TMX. The Resazurin reduction assay was used to evaluate the TMX cytotoxicity on MCF-7 breast cancer cell line and PBMCs human. The PCL-TMX nanofiber was cytotoxic to PBMCs and MCF-7. Based on these results, the PCL-TMX nanofibers developed have potential as an alternative for local chronic TMX use for breast cancer treatment, however tissue tests must be done.

## INTRODUCTION

Breast cancer is the second leading cause of death in women, after lung cancer (*Sung et al., 2021*). Treatment strategies for this disease include surgery, radiation therapy, chemotherapy, hormonal therapy, and targeted therapy, often applied in combination. Endocrine therapy is the treatment of choice for patients with hormone receptor–positive (HR+) breast cancer.

Breast cancer subtypes are defined by the expression of estrogen (ER) and progesterone (PR) receptors and the status of the HER-2 gene, which may be amplified. Breast cancer

cell lines with molecular profiles similar to those of tumors are used to evaluate the effects of anticancer drugs in vitro. The MCF-7 cell line derives from the pleural effusion of a patient with breast adenocarcinoma and represents the luminal A breast cancer subtype because it has the same molecular profile (ER+, PR+, normal HER-2 status). The use of anti-ER drugs, such as tamoxifen (TMX), for the treatment of HR+ (ER+ and PR+) breast cancers is widely accepted (*Johnston et al., 2016*).

TMX, also known as 4-hydroxytamoxifen, is a nonsteroidal compound that selectively modulates the ER with antagonistic or agonist action, depending on the organ on which it acts (*Salami & Karami-Tehrani, 2003*). It is an agonist in the liver, uterus, and bones, and an antagonist in the brain and mammary glands and vasomotor symptoms (*Sestak et al., 2006*). TMX has been used to stop the proliferation and inducing apoptosis of breast tumor cells through its anti-ER action (*Mandlekar & Kong, 2001*; *Salami & Karami-Tehrani, 2003*).

TMX has cytostatic and cytotoxic properties in the MCF-7 breast cancer cell line, not only stopping proliferation and inducing apoptosis, but also inducing differentiation and reducing cholesterol synthesis (*Kedjouar et al., 2004*; *Medina, Favre & Poirot, 2004*); it also modulates immunity in patients with breast cancer (*Robinson et al., 1993*; *Behjati & Frank, 2009*). Compared with those of healthy controls, lymphocytes from women with breast cancer treated with TMX showed significantly reduced killer activity, associated with a decrease in the absolute number of CD4-type lymphocytes, and a greater proliferation response in the presence of the concanavalin A mitogen (*Rotstein et al., 1988*; *Robinson et al., 1993*; *Behjati & Frank, 2009*). TMX is also effective against Ebola virus (*De Clercq, 2015*) and human immunodeficiency virus (*Laurence, Cooke & Sikder, 1990*) infections, and a recent review highlighted its benefits in the treatment of respiratory diseases, such as coronavirus disease pneumonia (*Salman Almosawey et al., 2020*). TMX and its active metabolites have prolonged serum half-lives, and higher doses have not been associated with improved outcomes; lower dosages have not been tested adequately. Furthermore, it is established in the literature that chronic TMX use (for ≥5 years) could reduce the risk of death from PR+ ER+ breast cancer (*Salami & Karami-Tehrani, 2003*; *Karn et al., 2010*; *Group (EBCTCG), 2011*; *Hong et al., 2016*; *Drăgănescu & Carmocan, 2017*) but when is used with potent inhibitors of CYP2D6 could be a risk of mortality (*Donneyong et al., 2016*).

It is well known that the nanometric scale devices used in current research for the prevention, treatment and diagnosis of diseases such as cancer are mostly natural or synthetic polymers. The desirable properties for these materials are biocompatibility, biodegradability, allowable controlled release of active agents and similarity to the native extracellular matrix of human tissues and cells (*Venugopal, Zhang & Ramakrishna, 2005*; *Caracciolo et al., 2011*; *Rogina, 2014*; *Mochane et al., 2019*). At the same time, local delivery systems have the advantage over systemic therapy of continuous drug delivery at higher concentrations directly to target sites. The benefits of these systems include improved patient compliance, the reduction of toxic effects and systemic complications (*Vyas, Sihorkar & Mishra, 2000*; *Jain et al., 2008*; *Joshi et al., 2016*), mimicking of the natural extracellular matrix (ECM), and highly efficient drug loading for controlled release.

Nanoscale systems, such as those in which nanofibers are employed, may present a promising opportunity for the efficient treatment of solid tumors.

Electrospun nanofibers are nanometric structures produced with synthetic or natural elements that create continuous filaments with a maximum diameter of 500 nm (*Caracciolo et al., 2011*). These materials are preferably biodegradable, to avoid the use of additional systems for their removal. The simplest use of nanofibers as a local drug release system involves the preparation of a polymer solution and its mixing with the drug, followed by nanofiber manufacture. Among numerous methods, electrospinning is becoming the main technique for the production of materials and carpets made of nano-polymer fibers and metal oxide (*Barnes et al., 2007*; *Duque, Rodriguez & Lopez, 2013*; *Rogina, 2014*; *Li et al., 2019*). This method is simple, versatile, common, and economical (*Rogina, 2014*); it is performed in an electrospinning machine, which enables the use of different compounds and control of manufacturing parameters to determine the diameter, size, and porosity of the continuous nanofibers produced (*Barnes et al., 2007*). Various biopolymers have been used for tailored biomedical applications (*Mochane et al., 2019*).

Most nanometric-scale devices used in current research on the prevention, diagnosis, and treatment of diseases such as cancer are made of natural or synthetic polymers. Desirable properties for these materials are biocompatibility, biodegradability, capacity for controlled release of active agents, and similarity to the native ECM of human tissues and cells (*Venugopal, Zhang & Ramakrishna, 2005*; *Caracciolo et al., 2011*; *Rogina, 2014*; *Mochane et al., 2019*). Poly ($\varepsilon$-caprolactone) (PCL), a semi-crystalline aliphatic polyester, is the most commonly used synthetic polymer in medical applications because it biodegrades slowly and is biocompatible, given its similarity to natural tissue components such as collagen fibers and ECM and diameters of 50–500 nm (*Venugopal, Zhang & Ramakrishna, 2005*; *Barnes et al., 2007*). PCL has good mechanical properties and thermal stability and is easy to process, compatible with hard and soft tissues, and accepted by the US Food and Drug Administration as a drug-delivery vehicle (*Song et al., 2018*). It has been used to develop devices for anticancer molecule release, an emerging promising alternative for cancer treatment (*Monteiro et al., 2017*).

In the present study, we developed PCL-pristine (P) and PCL-TMX nanofibers by electrospinning (*Vitchuli et al., 2011*) for local drug delivery to solid breast tumors. The nanofibers were characterized by scanning electron microscopy (SEM), Fourier-transform infrared spectroscopy (FTIR) and attenuated total reflectance infrared spectroscopy (FTIR-ATR), X-ray powder diffraction (XRD), thermal analysis, and contact angle measurement. Resazurin assays (*Escobar, 2010*) were used to assess their cytotoxic effects on MCF-7 cells and peripheral-blood mononuclear cells (PBMCs) from a healthy donor.

## MATERIALS & METHODS

### Materials

PCL with molecular weights of 43,000–500,000 was purchased from Polysciences, Inc. (USA). TMX was supplied by Araujo Drug Supply S.A. (Brazil). Dichloromethane (DCM; 99.5%) and methanol (MetOH; 99.8%) were acquired from Vetec (Brazil). All reagents were of analytical grade and were used as received.

## Preparation of PCL nanofiber solutions and electrospinning

To prepare the PCL-P and PCL-TMX polymer solutions, 800 mg PCL was dissolved in 10 mL DCM/MetOH mixture (50%/50% v/v) in each case. For the PCL-TMX solution, 15 mg TMX was added. The solutions were agitated for 12 h at 25.0 °C before use.

Each polymer solution was loaded into a 10-mL standard plastic syringe fitted with a 27-G blunted stainless-steel needle using a syringe pump (PHD 2000; Harvard Apparatus). The distance between the needle and the aluminum foil–wrapped collector was set at 15 cm, and electrospinning was performed with a solution flow rate of 10 mL/h and voltage of 20 kV generated by a high-voltage power supply (Gamma High Voltage, USA). The resulting nanofibers were collected and stored for physicochemical characterization and microbiological and cytotoxicity testing.

## Physicochemical characterization

The conditions for the physicochemical characterization of the PCL and PCL-TMX nanofibers were similar and adjusted according to earlier studies (*Ramírez-Agudelo et al., 2018*; *Dias et al., 2019*).

Nanofiber morphology was analyzed by scanning electron microscopy (SEM) (FEG-Quanta 200; FEI) with an accelerating voltage of 20 kV. Before analysis of SEM images, each nanofiber sample was coated with a 5-nm-thick layer of gold using a sputter coater (MD20; Bal-Tec). The average nanofiber diameter was calculated from at least 100 measurements obtained with Image J software (National Institutes of Health, Bethesda, USA). ATR was performed with a spectrophotometer (Spectrum 1SR; Perkin Elmer) equipped with a universal ATR sampling accessory and a diamond top plate. The FTIR-ATR spectra of the PCL-P and PCL-TMX nanofibers and TMX were obtained in the region of $4000–650 \, cm^{-1}$, with four scans obtained per sample at a resolution of $4 \, cm^{-1}$. The data were analyzed with the Spectrum software provided with the instrument (Perkin Elmer). The XRD patterns of the nanofibers were visualized using an X-ray diffractometer (XRD-7000; Shimadzu) with Cu $K\alpha\lambda = 0.154051$ radiations over a $2\theta$ range of $4–60°$ at a scanning speed of $2\theta$/min. Thermogravimetric and differential thermogravimetric analysis (TGA/DTG) was performed using a TGA Q5000 device (TA Instruments, USA) with sample heating at a rate of 10 °C/min from 25 °C to 600 °C and under an $N_2$ flow rate of 50 mL/min. The TG curves represent the thermal degradation of the samples. The data were processed using the software supplied with the instrument (Universal Analysis 200; TA Instruments).

A contact angle measuring system (SEO Phoenix 300 Touch) was used to determine nanofiber wettability. The nanofibers were placed on a sample stand, and water was dropped onto their surfaces while a camera recorded an image. The Surfaceware 9 software was used to determine the average contact angle.

To evaluate the drug release profile of the PCL-TMX nanofibers (loaded at 25.5 µg TMX/mg nanofiber approximately), approximately 15 mg of nanofibers was cut into specimens (15 × 15 mm), which were placed into Eppendorf tubes. The tubes were then incubated at 37 °C in 2 mL phosphate-buffered saline (PBS; pH 7.4) with 0.01% Sodium Dodecyl Sulfate (SDS) for increasing the TMX solubility, in a thermostatic shaker at 50 rpm. Samples (2 mL) were removed at 0.5, 1, 2, 4, 8, 10, 24, 48, 72, 120, and 144 h for

the quantification of TMX release; after each analysis, the same volume of fresh PBS solution was added to the tube. The amount of TMX released was determined using an ultraviolet–visible (UV-vis) spectrophotometer (Multiskan Spectrum MCC/340; Thermo Scientific) at a wavelength of 365 nm, based on a calibration curve ($R^2 = 0.99$). Each sample was evaluated in triplicate.

## Cytotoxicity testing

For the *in vitro* analysis of cytotoxicity, the MCF-7 cell line was obtained from frozen vials of laboratory stock obtained from the ATCC (Manassas, VA, USA). The MCF-7 cells were grown in Dulbecco's modified Eagle medium (DMEM; Gibco) prepared with 1% (v/v) antibiotic and antimycotic solution (ref. 15240062; Gibco) and supplemented with 10% fetal bovine serum (FBS; Gibco), in 75-cm$^2$ plastic bottles at 37 °C in a 95% humid atmosphere with 5% $CO_2$.

To avoid the interference in the experiment of the action of the steroids present in the FBS, and of the weak estrogenic activity of the phenol red present in the DMEM, the cells were washed with PBS and then medium with 10% carbon-stripped FBS (Sigma) and phenol red–free DMEM was added 48 h before incubation with the nanofibers.

For peripheral-blood mononuclear cell (PBMCs) isolation, 10-mL peripheral blood samples were obtained by venipuncture of the brachial vein from a healthy volunteer who had provided informed consent. The blood was collected into tubes with heparin, and PBMCs were obtained using a Ficoll gradient procedure (*Rotstein et al., 1988*). Briefly, the 10-mL tubes of blood were centrifuged at 2,000 rpm for 5 min, and the buffy coat was then removed with a sterile 2-mL pipette. The buffy coat (2 mL) was added gently to a 15-mL tube with 2 mL Ficoll Histopaque-1077 (Sigma), which was centrifuged without brake for 20 min at 2,000 rpm. Then, the white layer was recovered with a sterile 2-mL pipette and placed in a new tube with 5 mL PBX 1X prepared from a 10X solution (ref. 70011044; Gibco), which was centrifuged twice at 2,500 rpm for 5 min. The cell pellet was then recovered, gently resuspended, and placed in a new 15-mL tube containing 5 mL PB-MAX karyotyping medium (Invitrogen) with 100 μL phytohemagglutinin (PHA) M (ref. 10576-015; Gibco) and antibiotic and antimycotic solution (ref. 15240062; Gibco). The tube was stored at 37 °C and 5% $CO_2$ for 24 h before nanofibers treatment.

The cytotoxicity activity of PCL-TMX nanofibers were evaluated by an indirect contact resazurin assay. The mean absorbance values obtained for all groups were distributed normally, and the control group data were adjusted to 100% viability. Cytotoxicity was calculated based on cell viability relative to this group: none, >90%; slight, 60–90%; moderate, 30–59%; and severe, <30% (*Basak et al., 2016*). This test indicates the number of viable cells and the level of metabolic activity in a sample. Resazurin, a blue dye, is metabolized by mitochondrial enzymes in cells, which transforms it into fluorescent pink resorufin, which the cells release into the culture medium. Treatments can be monitored by taking several measurements of the same group of cells, as resazurin is not toxic. Plates were removed from the incubator for a short time (5–10 min) to take measurements, and the culture conditions were then restored (*Escobar, 2010*; *Uzarski et al., 2017*). The MCF-7 cells were seeded in a 96-well culture plate at a density of 15,000/200 μL for 24 h. Then,

a seven mm diameter circle containing approximately 16 µM of TMX was added to each well with 200 µL the culture medium for 1–6 days, in duplicate. Every day, a plate was taken from the incubator and the culture medium was removed; the wells were washed with 200 µL PBS, and fresh serum-free medium with 4.4 µM resazurin was added, followed by further incubation under the same initial conditions. After 4, 6, and 24 h, absorbance was measured at the emission wavelength of 595 nm and excitation wavelength of 535 nm using a spectrophotometer (Cytation 3 (Borra et al., 2009; *Uzarski et al., 2017*).

An experiment with free tamoxifen was made, MCF-7 cells were cultured with concentrations of free TMX between (0–20µM), an effect similar with viability percentage to PCL-TMX, was observed with concentrations between 13 and 20 µM (see Fig. S1).

For PBMCs assays, 15,000 cells/200 µL were placed in 96-well plates after 24 h culture and incubated with the PCL-P and PCL-TMX nanofibers, for 24 h. The plates were then centrifuged at 2,000 rpm for 5 min, and the medium was replaced with PB-MAX containing 4.4 mM resazurin, followed by further incubation under the same conditions. After 4 h, 6 h, 24 h, 30 h and 48 h the absorbance was measured the same way as with MCF-7 cells lines.

## Statistical analysis

The results were organized by treatment: PCL-P, PCL-TMX and without nanofiber and by numbers of resorufin measurements according to cell type. The normality test Shapiro–Wilk was applied, then the treatments were compared using Student's t test for unpaired variables: MCF-7+PCL-P *vs.* MCF-7+PCL-TMX, MCF-7+PCL-P *vs.* MCF-7 without nanofiber, MCF-7+PCL-TMX *vs.* MCF-7 without nanofiber and PBMCs+PCL-P *vs.* PBMCs+PCL-TMX, PBMCs+PCL-P *vs.* PBMCs without nanofiber, PBMCs +PCL-TMX *vs.* PBMCs without nanofiber). Here we found that the PCL-TMX treatment reduced the percentage viability of MCF-7, and the difference was statistically significant in all measurements ($p \leq 0.05$). One day after PCL-P treatment of MCF-7 cells, the percentage of viability increases slightly and it is statistically significant only in first day at 4 h ($p = 0.0160$) and on the sixth day at 24 h ($p = 0.0317$), in the latter case, a clone of MCF-7 with a higher proliferation rate probably emerged. PCL-P induced an increase in the percentage of viability of PBMCs on the first day of treatment, even above the cells without treatment, while PCL-TMX reduced it however, applying the Student's *t* test the differences were not significant in the PCL-P treatments, but they were significant for the PCL-TMX at 4 h $p = 0.005$ and 6 h $p = 0.0243$.

Also, ANOVA was applied to three treatments for each cell type (MCF-7+PCL-P *vs.* MCF7+PCL-TMX *vs.* MCF-7 without nanofiber and PBMCs +PCL-P *vs.* PBMCs+PCL-TMX *vs.* PBMCs without nanofiber). All treatments showed statistically significant differences, but this test does not discriminate between groups. Additionally, BONFERRONI test was used to compare each treatment with no nanofiber addition (MCF-7+PCL-P *vs* MCF-7 without nanofiber, MCF7+PCL-TMX *vs* MCF-7 without nanofiber and PBMCs+PCL-P *vs* PBMCs without nanofiber, PBMCs+PCL-TMX *vs.* PBMCs without nanofiber). This showed that PCL-P increases the percentage of viability in a statistically

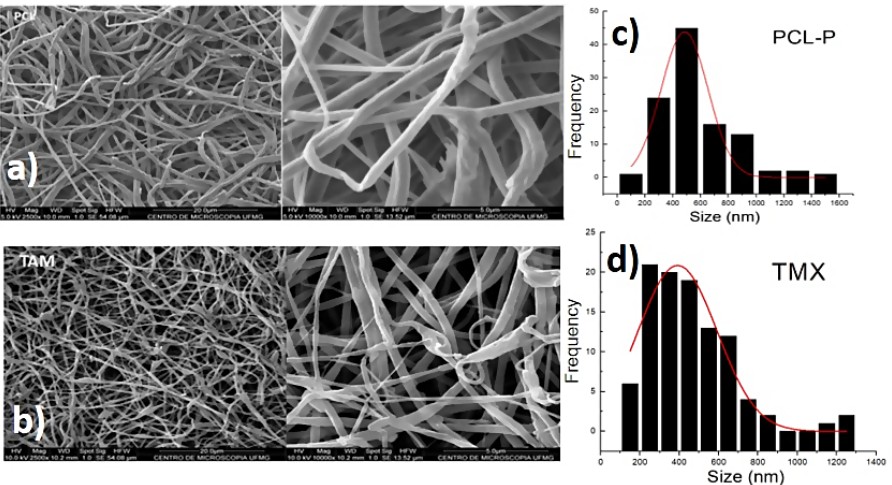

**Figure 1** **Morphology and diameter distribution.** The morphology and diameter distribution of PCL nanofibers showed in micrographs, and histograms corresponding of PCL nanofibers SEM micrographs of (A) PCL-P, (B) PCL-TMX. Histograms of the diameter of nanofibers of (C) PCL-P, (D) PCL-TMX.

significant way except for the 6 h data. On the other hand, PCL-TMX decreases this value and only in a statistically significant way for the measurement taken at 24 h.

# RESULTS AND DISCUSSION

## Physicochemical characterization

SEM showed that all PCL-P and PCL-TMX nanofibers were uniaxial, non-porous, and distributed randomly. The incorporation of TMX altered the nanofiber morphology and diameter (see Fig. 1). The PCL-P nanofibers displayed a bimodal diameter distribution, whereas that of the PCL-TMX nanofibers was modal. The largest average diameters were 484 ± 168 nm for PCL-P nanofibers and 400 ± 236 nm for PCL-TMX nanofibers. The PCL-P nanofiber diameters were comparable to those reported previously (*Katsogiannis, Vladisavljević & Georgiadou, 2015*), and the reduction of the average diameter with TMX incorporation is consistent with previous reports that drug or particle incorporation reduces PCL nanofiber diameters (*Zamani et al., 2010*; *Aristilde et al., 2010*; *Monteiro et al., 2017*; *Alavarse et al., 2017*; *Pinzón-García et al., 2017*).

The FTIR-ATR spectra of TMX and the PCL-P and PCL-TMX nanofibers are shown in Fig. 2. TMX showed a band of intensity at 3,229 cm$^{-1}$ due to the O–H from alcohol and phenolic groups. The most characteristic TMX bands were observed: the C = O band at 1,627 cm$^{-1}$, the N–H band at 1,575 cm$^{-1}$, the C = C stretching band (reflecting aromatic ring vibrations) at 1,453 cm$^{-1}$, the double amino C–N stretching bands at 1,227 cm$^{-1}$, and the phenolic C–O stretching band at 1,174 cm$^{-1}$ (*Aristilde et al., 2010*; *Dos Santos Ferreira da Silva et al., 2015*). For the PCL-P nanofiber, characteristic infrared bands were observed at 1,720 cm$^{-1}$ (C = O carbonyl stretching), 1,240 cm$^{-1}$ (asymmetrical C–O–C stretching), 1,157 cm$^{-1}$ (symmetrical C–O–C stretching), 2,945 cm$^{-1}$ (asymmetrical CH$_2$ stretching), and 2,868 cm$^{-1}$ (symmetrical CH$_2$ stretching) (*Elzein et al., 2004*; *Gomes et al., 2008*). For
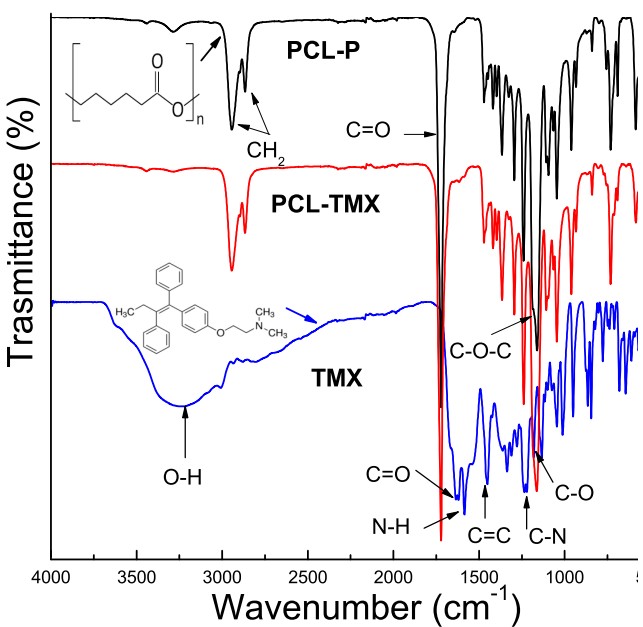

**Figure 2** FTIR spectra of TMX, PCL-P, and PCL-TMX electrospun nanofibers.

the PCL-TMX nanofibers, the TMX absorption peaks were not observed, likely due to the small quantity of TMX in them and stretching overlap with adsorption bands of the PCL polymer (*Liebenberg et al., 1999*; *Khalf & Madihally, 2017*).

Figure 3 shows the XRD patterns of TMX and the PCL-P and PCL-TMX nanofibers. TMX showed low-intensity peaks due to its polycrystalline structure; the main peaks were at $2\theta = 8.5°, 9.3°, 10.6°, 17.0°, 21.1°$, and $23.0°$ (*Liebenberg et al., 1999*; *Thangadurai et al., 2005*; *Toro et al., 2007*). All PCL-P nanofibers showed two characteristic peaks at $2\theta = 21.25°$ and $23.55°$, attributed to (110) and (200) PCL semicrystalline lattice planes (*Baji et al., 2007*; *Wang, Guo & Cheng, 2008*; *Kim et al., 2012*). No characteristic TMX peak was detected in the PCL-TMX nanofiber pattern. These results can be explained by the lack of time for the polymers and other compounds to crystallize and form organized structures during electrospinning, which is a very rapid method of polymer fiber preparation (*Wei et al., 2010*).

Figure 4 shows the TG and DTG curves of TMX and nanofibers. These curves for free TMX showed that three events of mass loss occurred at temperatures of up to 600 °C. The first event corresponds to 7.4% mass loss at about 100 °C, attributed to dehydration. The second event occurred in two steps (first between 126 °C and 200 °C with 20% mass loss, and second between 200 °C and 280 °C with 50% mass loss, also observed on the DTG curve), and the third event occurred between 500 °C and 600 °C, with 94% mass loss. These events can be attributed to the oxidative decomposition of TMX and the remaining carbonaceous matter (*Dos Santos Ferreira da Silva et al., 2015*; *Cervini et al., 2015*). The PCL-TMX nanofiber patterns (red line) were similar to the reported PCL weight loss pattern (black line), which comprises three events of mass loss, the most critical thermal

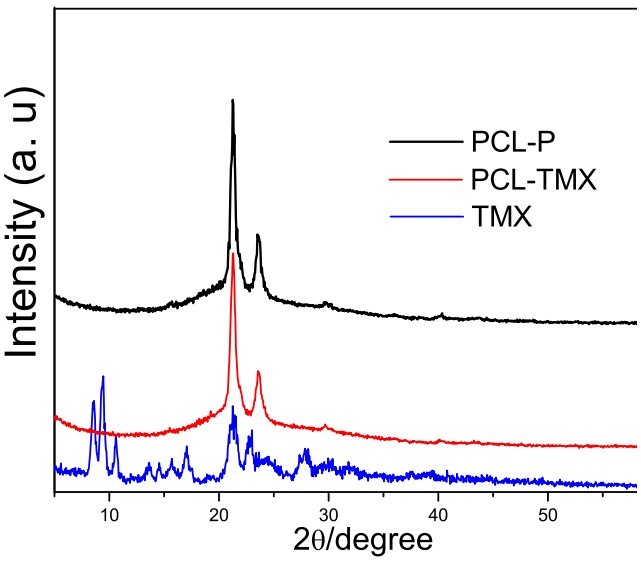

**Figure 3** **XRD patterns of (A) TMX, (B) PCL-P, (C) PCL-TMX.**

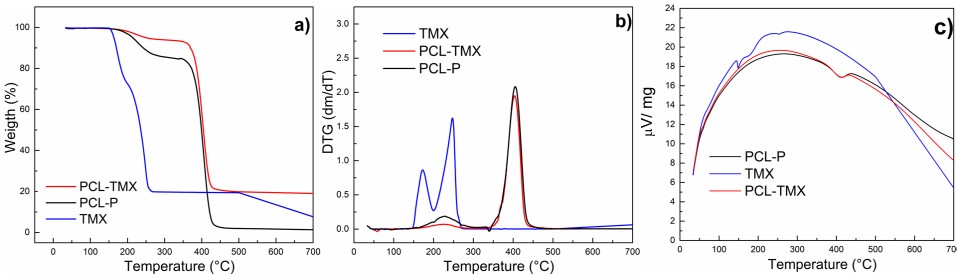

**Figure 4** **Thermogravimetric analysis.** (A) TG, (B) DTG and (C) temperature difference curves of TMX, PCL-P, and PCL-TMX.

decomposition event starting at about 340 °C and ending at 470 °C (*Mohamed et al., 2008*) with 70% mass loss for PCL-TMX and with total weight loss for PCL-P at 600 °C. In this way, the PCL-TMX nanofibers showed more excellent thermal stability than did the PCL-P nanofibers and free TMX.

The contact angles of the nanofibers surfaces were measured to assess the wettability and hydrophilicity of the nanofibers. Table 1 shows the contact angles for the PCL nanofibers and PCL-TMX nanofibers. The contact angle of the PCL-TMX nanofiber was smaller than that of the PCL-P nanofiber (hydrophobic nature (*Madhaiyan et al., 2013*; *Tiyek et al., 2019*)), perhaps due to the highly hydrophilic COO–moiety of citrate TMX on the surface of the former (*Huang et al., 2010*). Greater nanofiber wettability may improve cell proliferation and biocompatibility (*Sharma et al., 2014*). In a similar work, the incorporation of 5-FLU, paclitaxel, and other drugs into PCL nanofibers also increased

Table 1  Contact angle of the PCL nanofibers after 1s and 120s.

| Nanofibers | Time/s | Contact angle/degrees |
| --- | --- | --- |
| PCL-P | 1 | 105.75 |
| | 120 | 96.58 |
| PCL-TMX | 1 | 55.79 |
| | 120 | 32.54 |

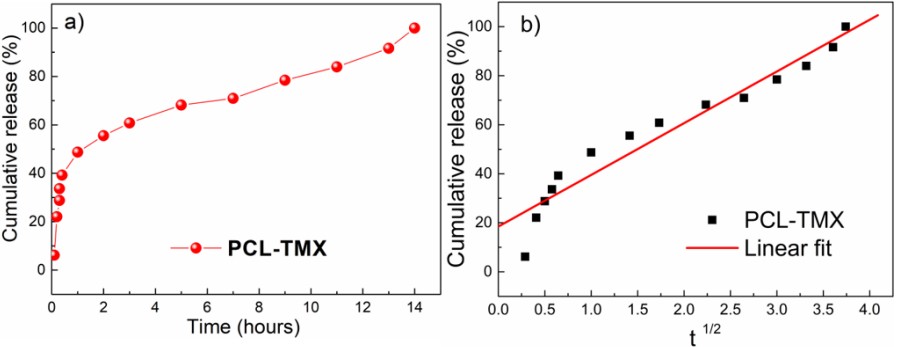

**Figure 5** **Drug release.** (A) Release profiles of TMX from PCL-TMX nanofibers in PBS pH 7.4 and (B) Higuchi equation to TMX release from PCL-TMX nanofiber (Where $M_t/M_\infty$ is the fraction of TMX delivery in time t, and K is release speed constant).

nanofiber hydrophilicity and provided a good release profile (*Karuppuswamy et al., 2015*; *Iqbal et al., 2017*).

### *In vitro* drug release

Profiles of cumulative TMX release from the PCL-TMX nanofibers over 14 h are shown in Fig. 5. After 12 h, no TMX signal was detected by UV-vis quantification, reflecting complete TMX release. Thus, to evaluate the kinetics (linear fit) of TMX release from the nanofibers, the cumulative release of the drug was considered between 1 h to 12 h. The large surface areas and three-dimensional open porous structures of nanofibers may reduce the constraint on drug diffusion and release (*Ramakrishna, Zamani & Prabhakaran, 2013*). In addition, the greater hydrophilicity of PCL-TMX nanofibers certainly increased TMX release.

Three distinct, sequential stages of TMX release, reflecting different diffusion processes from the PCL matrix, were observed (Fig. 5A), following the literature (*Jannesari et al. 2011*; *Sohrabi et al., 2013*). In the first stage, there was a linear relationship with a very pronounced and moderate slope. A burst effect was seen, with approximately 50% of the TMX release in the first hour of the experiment (first 6 points). This initial rapid release may have been due to the accumulation of the TMX molecules at or near the PCL nanofiber surfaces during electrospinning, facilitating TMX release into the media (*Zamani et al., 2010*). The second stage of release occurred between 1 and 4 h; the release rate decreased gradually, resulting in a moderate slope resulting from TMX diffusion through the PCL nanofibers instead of PCL degradation with the release of approximately 70% of the TMX.

The third stage occurred between 4 and 10 h and involved the most negligible TMX release that reaches more than 87% of the TMX release. Thus, sustained TMX release from the PCL-TMX nanofibers was observed up to 14 h. These findings are expected for this type of system because an initial burst of drug release is required to promote a local antitumor effect; the initial dose kills most cancerous cells, and the subsequent controlled release prevents tumor cell growth and proliferation (*Ma et al., 2011*).

The mechanism of TMX release was evaluated using the Higuchi kinetic model, based on Fickian diffusion mechanism (*Nie et al., 2009*). The Higuchi model of the TMX release mechanism best fit to the data for the first 8 h (Fig. 5B). As this model assumes Fickian diffusion, the cumulative percentage of the drug released ($Q$) was plotted against the square root of time ($t\frac{1}{2}$), *i.e.*, $Q = K \times t\frac{1}{2}$, where $K$ is the Higuchi rate constant. The results indicated that diffusion along the PCL matrix occurred, and that TMX release was not dominated by polymer erosion, as claimed in previous studies of biodegradable polymers and water-soluble molecules (*Luong-Van et al., 2006*; *Fredenberg et al., 2011*).

Other studies of drug-delivery systems for anticancer molecules, such as TMX, have shown sustained release over 10 h (*Guimarães et al., 2015*), 6 days (*Criado-Gonzalez et al., 2019*), 8 days (*Liu et al., 2016*), 14 days, 25 days (*Iqbal et al., 2017*), and 35 days, with different release mechanisms and cytotoxicity. Formulations of nanofibers loaded with tetracycline hydrochloride (TCH), an antibiotic in the same group as TMX, have shown good cytocompatibility in normal cells (*Qi et al., 2013*; *Ranjbar-Mohammadi et al., 2016*; *Alavarse et al., 2017*). Similarly, in this study, PCL-P nanofibers showed good biocompatibility, making it a potential candidate for use as a TMX local delivery system since the cytotoxic effect was only observed when the PCL-TMX nanofiber was present.

## Cytotoxicity

All treatments (MCF-7+PCL-P and PCL-TMX, PBMCs+PCL-P and PCL-TMX) showed statistically significant differences (Figs. 6 and 7) by ANOVA test. In order to discriminate between groups, we used Student's $t$-test and each treatment was compared against control (PCL-TMX *vs* no nanofiber and PCL-P *vs* no nanofiber). After, 24 h PCL-TMX showed more significant cytotoxicity against MCF-7 than did the PCL–P; 20% ($p = 0.041$) cell viability was observed on the first day, in contrast to the 100% ($p = 0.049$) and 127% ($p = 0.016$) viability observed in untreated cells and those incubated with PCL-P, respectively. After the second day, resazurin metabolism was barely detected in cells incubated with PCL-TMX; those incubated with PCL-P showed a slight decrease in viability.

This can be explained because the TMX inhibits MCF-7 proliferation (*Niro, Hennebert & Morfin, 2010*), arresting cells in the G0–G1 phases of the cell cycle. It also activates apoptosis *via* procaspase 8, followed by events such as an increase in reactive oxygen species and the release of pro-apoptotic factors from the mitochondria. Real-time polymerase chain reaction revealed an increase in FasL mRNA and tumor necrosis factor- $\alpha$, as well as a decrease in mitochondrial transmembrane potential, after TMX treatment. All these changes are related to the activation of apoptosis (*Subramani et al., 2014*). After 48 h of PCL-TMX incubation, a 99% ($p = 0.016$) reduction in cell viability, representing a

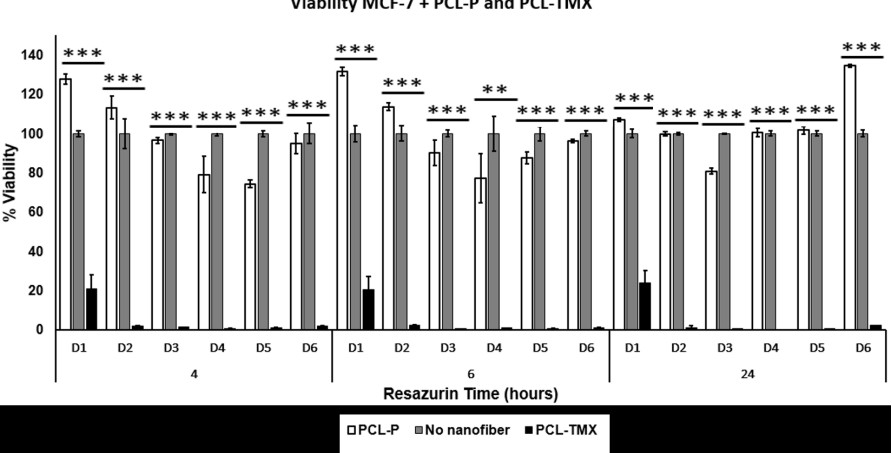

**Figure 6 Cytotoxicity assay.** Percentage of MCF-7 cell viability: cytotoxic effect of PCL-P and PCL-TMX on 15.000 MCF-7 cells was evaluated after 1 to 6 days of exposure. The reduction of resazurin to resorufin was followed at 4, 6, and 24 h. PCL-TMX statistically significantly reduced the percentage of viability by the ANOVA method in all treatments.

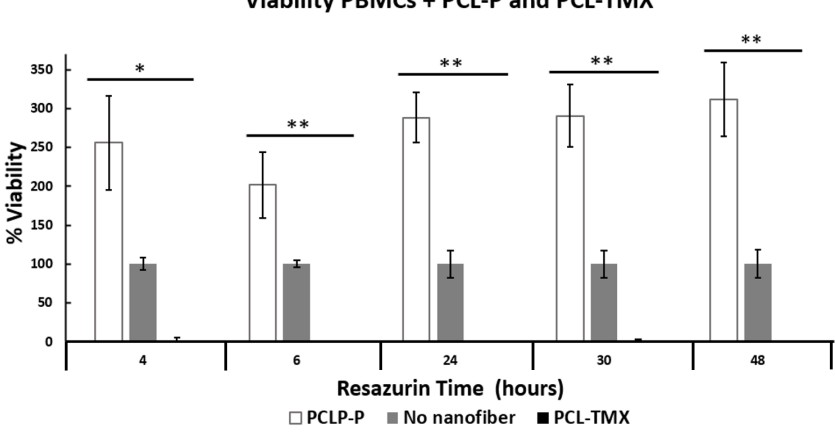

**Figure 7 Percentage of PBMCs viability.** Percentage of PBMCs viability: cytotoxic effect of PCL-P and PCL-TMX on cultured 15.000 PBMCs for 24 h. The reduction of resazurin to resorufin was followed up at 4, 6, 24, 30 and 48 h. Increase in the percentage of viability was observed with exposure to PCL-P and reduction with PCL-TMX. The level of significance obtained by the ANOVA method is indicated with asterisks, the values are in the Table S1.

significant difference from the other groups, (Table S1). On day 6, slight recovery with 2% viability was observed, possibly reflecting the growth of a treatment-resistant clone. When MCF-7 cells were cultured with concentrations of free TMX in the range of 0–20 μM, a similar effect on viability percentage to PCL-TMX, was observed with concentrations between 13 and 20 μM.

Also, TMX drastically reduced the viability of PBMCs obtained from healthy donors, when PCL-TMX added to the culture medium, with statistical significance based on the

ANOVA method. After the first day of exposure, the viability percentage reached zero at 4 h ($p = 0.0126$) and 6 h ($p = 0.089$). On the contrary, PCL-P increased in viability to 256%, which shows that this polymer could be activated PBMCs proliferation (Fig. 7). In PBMCs obtained from breast cancer patients, the effect of TMX is different depending on the duration of treatment. For example in a short time, showed no change in cytotoxic activity type natural killer (NK) cells or proliferative response to mitogens after 8 days of TMX treatment (*Sheard et al., 1986*). This means that TMX does not activate the NK cells from PBMCs nor induce their proliferation. In a similar study involving 6–12 months of treatment, a reduced number of lymphocytes with suppressive function was observed (*Joensuu, Toivanen & Nordman, 1986*). In patients treated with TMX for 1.5–2 years, a decrease in NK cell activity and increase in response to the concanavalin A mitogen were observed (*Mandeville, Ghali & Chausseau, 1984*). This suggests that if TMX is applied locally, unwanted effects on other tissues can be avoided.

For PBMCs obtained from the peripheral blood of patients with breast cancer and treated with TMX or left untreated, the response to concanavalin A can take up to 5 days (*Rotstein et al., 1988*). In this study, PBMCs obtained from the peripheral blood of a healthy volunteer and cultured in the presence of PHA showed detectable metabolic activity in the resazurin assay until the fourth day of culture (Fig. 7). However, lymphocytes in culture under the stimulation of a mitogen such as PHA are viable for about 72 h. The viability of PBMCs increased almost threefold in the presence of PCL-P and decreased by approximately 99% with of PCL-TMX. Little is known about the possible activation and proliferation of PBMCs induced by PCL-P, however they have been used to promote tissue healing in order to promote cell migration (*Schoenenberger et al., 2020*). In further research, it would be interesting to delve into this aspect. Viability reduction by PCL-TMX is consistent with that reported by *De Oliveira, Genari & Dolder (2010)*, who showed cell death due to apoptosis and autophagy in lymphocytes treated with tamoxifen for 24 and 48 h, in a time—dependent manner, although they applied a dose of 20 μM while in this study it was 16 μM. They conclude that the effect of TMX on lymphocytes is independent of the estrogen receptor (*Behjati & Frank, 2009*). Other side effects of TMX are the induction of proliferation in the endometrium, association with liver cancer, increased blood coagulation, retinopathy and corneal opacities formation (*Memisoglu-Bilensoy et al., 2005*). Among the potential biomedical (drug-delivery) applications of electrospun nanofibers, local postoperative chemotherapy for the prevention of tumor recurrence and metastasis is prominent (*Hu et al., 2014*). PCL-P is used to administer several types of drugs in the treatment of cancer such as cisplatin, doxycline, curcumin, paclitaxel among others (*Malikmammadov et al., 2018*), but there are no reports of tamoxifen delivery systems with PCL nanofibers under the conditions described here.

In this study, the PCL-P nanofibers displayed good biocompatibility, and thus potential application for release TMX. Similarly, nanofibers loaded with TCH have shown good cytocompatibility in normal cells (*Qi et al., 2013*; *Ranjbar-Mohammadi et al., 2016*; *Alavarse et al., 2017*). Other TMX release systems using nanoparticles achieve 68% release in a first hour (*Chawla & Amiji, 2002*) then maintains the release until 24 h, while nanofibers released up to approximately 50% of their content in the first time and maintain a sustained

release for hours or days, which can give best results. In addition, the destruction of the nanoparticles requires the use of enzymes such as lipases, which can affect the environment of the treated tissue.

## CONCLUSIONS

PCL-TMX nanofibers were produced effectively by electrospinning and showed sustained TMX release for up to 14 h. In cell viability assays, they exhibited excellent activity against the MCF-7 cell line. These results suggest that PCL-P nanofibers have potential application as a TMX-delivery local system (PCL-TMX), that could avoid the collateral effects of TMX treatment in other tissues such as endometrium, liver, cornea or cells such as PBMCs. There are no previous reports of PLC-TMX nanofiber releasing systems in the literature.

### Funding

This work was supported by Brazilian Research agencies: CAPES, FAPEMIG, CNPq and Center of Microscopy at UFMG, and the Universidad del Rosario and MinCiencias-COLCIENCIAS Contract: FP44842-07-2018 Code: 122277657905. The funders had no role in study design, data collection and analysis, decision to publish, or preparation of the manuscript.

### Grant Disclosures

The following grant information was disclosed by the authors:
Brazilian Research agencies: CAPES, FAPEMIG, CNPq and Center of Microscopy at UFMG, and the Universidad del Rosario and MinCiencias-COLCIENCIAS Contract: FP44842-07-2018 Code: 122277657905.

### Competing Interests

The authors declare there are no competing interests.

### Author Contributions

- Ana D. Pinzón-García conceived and designed the experiments, performed the experiments, analyzed the data, prepared figures and/or tables, authored or reviewed drafts of the paper, and approved the final draft.
- Ruben Sinisterra conceived and designed the experiments, performed the experiments, analyzed the data, authored or reviewed drafts of the paper, and approved the final draft.
- Maria Cortes conceived and designed the experiments, performed the experiments, authored or reviewed drafts of the paper, and approved the final draft.
- Fredy Mesa analyzed the data, prepared figures and/or tables, authored or reviewed drafts of the paper, and approved the final draft.
- Sandra Ramírez-Clavijo performed the experiments, prepared figures and/or tables, and approved the final draft.

## Data Availability

The raw measurements are available in the Supplemental File.

## Supplemental Information

Supplemental information for this article can be found online at http://dx.doi.org/10.7717/peerj.12124#supplemental-information.

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
