# Peer review of "Polycaprolactone nanofibers as an adjuvant strategy for Tamoxifen release and their cytotoxicity on breast cancer cells"

_PeerJ, doi:10.7717/peerj.12124_

## Round 0.1 · original submission · Major Revisions

Please consider engaging the services of a fluent English speaker or an agency offering editorial and proofreading services to correct the numerous grammatical errors in the manuscript.

Error bars should be included for the data points and bars in the plots, and in the table.

·

Basic reporting

a. The language in the article is generally clear and simple, but a thorough grammar check is needed throughout the manuscript to address issues that could cause confusion because of unclear language. Sentence structure should be revised as well. In general, the article text is technically correct.
b. The article is well cited and demonstrates good mentions of previous work and current understanding in the literature.
c. In general, the figures are well represented. However, figures 6 and 7 need to display error bars and sample size. It would be helpful to include the statistical analysis from the supplemental files directly on the graphs to point to statistical significance in the form of p-values (*). In figure 7, data are not presented clearly and could benefit from rearranging or even breaking up into 2 parts unless statistical tests are to be included comparing certain groups. The Raw data excel files need more annotation for better readability.

Experimental design

a. The article is well within the scope of the journal. The article clearly defines the research questions, and the experiments are proposed in attempt to answer the defined questions.
b. In general, the experiments and investigations are rigorous and are technically sound. The nanofiber material characterization is thorough and provides good insight on the effect of tamoxifen loading. Quantification of loading amount is needed. Good attempts to assess cytotoxicity for breast cancer cells. Results on nanofiber effects on lymphocyte cytotoxicity is not clearly presented or discussed. Information on amount of nanofibers dosed for cytotoxicity studies is needed.
c. The methods described in the article are clear and are repeatable. As mentioned above, dosing concentrations should be included in the methods.

Validity of the findings

a. In general, the data presented in the manuscript is robust and is a solid base for the conclusions. Statistical analyses are adequate, but more robust comparisons may be helpful including ANOVA tests for cytotoxicity studies and more discussion of statistical results. In general, experimental controls are present. For cytotoxicity experiments, free (bulk) Tamoxifen dosing at amounts comparable to nanofiber loading amounts are helpful as an additional control.
b. The conclusions from the studies are well presented and discussed.

Additional comments

1. Have you looked at range of concentrations for the nanofiber dosing? What amounts are being treated to cells?
2. What tamoxifen amounts are present in the nanofibers after loading? Is there a way to tune those amounts?
3. How are you ensuring that cells from buffy coat are purely lymphocytes? Flow cytometric analysis or potentially FACS may be helpful to ensure no contamination from other leukocytes that are usually present in the buffy coat from density gradient separations.
4. Figure 7 cell viability shows >250% cell viability; this needs more explanation and potentially different representation.

·

Basic reporting

A well written paper with Authors have reported use of tamoxifen citrate (TMX) loaded nanofibers for treatment of breast cancer. Following are some minor changes in reporting. Sufficient literature has been reviewed and cited. Data has been sufficiently reported in form of table and figures.

a. Some mistakes in sentence structures have been observed and should be fixed before final submissions.
b. Grammatic mistakes have been observed frequently in the manuscript.

Experimental design

The studies presents original results obtained from rigorous experimentation, keeping in view the technical and ethical standards. All methods have been described sufficiently. However, shortcomings of use of TMX (i-e the research gap) needs to be elaborated further.

Validity of the findings

Novelty of study lies in use of TMX, in conditions where it can't be used with conventional methods, thanks to the electrospinning process and nanofibers produced thereof. Underlying data has been provided adequately. The samples have been characterized to required level using a number of techniques that are well known in subjected domain. However, following observations were made:

a. The gap (i-e inappropriateness of TMX or shortcomings of TMX for which it was incorporated in nanofibers) has not been related to results. Authors reported interesting findings which need to be related to the research gap in more detail.
b. The conclusion section needs to be improved and linked to original research question.

Additional comments

The study presents an important application of electrospun nanofibers for treatment of breast cancer and seems worth publishing once the minor changes mentioned above have been addressed.

Reviewer 3 ·

Basic reporting

- The English needs review (I highlighted below a few examples of sentences with grammatical errors, but I found many others in the text).

- The introduction was too long. Also, it does not bring any information about the polycaprolactone nanofibers, in specific.

- The reason for choosing the polycaprolactone nanofibers as delivery material was not clear.

- The references should be updated. There are too many old references, some of them over 20 years of publication date.

Experimental design

- The paper follows the scope of the journal.

- I believe the authors presented well the importance of the nanofibers in general therapies, however, they should focus on the specific material they've chosen. Also, it would be interesting to show some data on the use of polycaprolactone nanofibers in other diseases, explaining the rationale why it must be a good option for TMX.

- The statistical analysis should be another subtopic.

Validity of the findings

- The statistical results were almost not presented in the text.

- The conclusions match the objectives of the study and the results are well stated.

Additional comments

- Abstract. The authors could make a better description of the in vivo results.

- Introduction. Lines 36-38. Where is the reference?

- Introduction. Lines 46-48. In the sentence "Anti-estrogens such as tamoxifen is widely accepted drug for the treatment of hormone receptor-positive (ER+ and PR+) breast cancers", the pronoun must be in the plural: "Anti-estrogens such as tamoxifen are widely accepted drugs for the treatment of hormone receptor-positive (ER+ and PR+) breast cancers".

- Introduction. Lines 52-53. The sentence "It is an agonist in the liver, uterus and bones, and as an antagonist in the brain and mammary gland." should be: ""It is an agonist in the liver, uterus and bones, and an antagonist in the brain and mammary gland."

- Introduction. Lines 76-78. Exclude the sentence "Nanotechnology in recent years has produced various kinds of nanodevices with applications in many fields; one of the most benefited has been that of medicine. In particular, the release of local medicines using controlled systems is of great interest."

- Conclusions. Lines 462-464. In the sentence "These results suggest that the PCL-TMX nanofibers have potential as a drug delivery system as adjuvant to treat solid tumors or breast cancer", I suggest trading "or" for "including".

---

## Round 0.2 · Minor Revisions

There are still grammatical and typographical errors to be corrected and inconsistencies between the text discussions and the figures need to be addressed. Please refer to the annotations in the attached PDF file.

·

Basic reporting

The language has been improved. Sufficient literature has been cited.

Experimental design

Experimental design and its execution is as per standard research requirements. Research question has been defined and answered in a better way compared to previous version.

Validity of the findings

The validity of results seems to be acceptable.

Additional comments

The manuscript may be proceeded for publication.

---

## Round 0.3 · Minor Revisions

1. There is a discrepancy between a statement in the Abstract and Figs 6 and 7. In the Abstract, the authors state "The PCL-TMX nanofiber was slightly cytotoxic in PBMCs and highly toxic in the MCF-7. Based on these results, the PCL-TMX nanofibers developed have potential as an alternative for chronic TMX use for breast cancer treatment without affecting other cells or tissues". Fig. 7 B, however, shows that PCL-TMS is extremely toxic to PBMC (indeed more toxic towards them than towards MDF-7 (cf. Fig. 6)). The authors must resolve this discrepancy.

2. The authors have made the necessary grammatical corrections. There are 2 minor corrections remaining:
(i) Line 444 (WORD document) - should be "After 24 h," instead of "Afterward, 24 h"
(ii) Line 469-470 (WORD document) - the word "when" is misplaced. The sentence should be "When MCF-7 cells were cultured with concentrations of free TMX in the range of 0-20M, a similar effect on viability percentage...."

---

## Round 0.4 · Minor Revisions

1. The changed sentence in the Abstract "The PCL-TMX nanofiber was cytotoxic in PBMCs and MCF-7" should be "The PCL-TMX nanofiber was cytotoxic to PBMCs and MCF-7".

2. The section editor has made additional comments:

(i) Line 430: "The effect of TMX on PBMCs is not clear. Here we observed a large reduction of viability." These sentences need to be clarified in the text. Does the first sentence "The effect of TMX on PBMCs is not clear" refer to past studies in the literature since the subsequent sentence and Fig 7b clearly show that TMX in the PCL-TMX is highly cytotoxic to PBMCs?

(ii) The y-axis and x-axis lines in Fig. 7 should be drawn in like Fig. 6

(ii) Why are the 3 sets of data in Fig. 7b for 24, 30 and 4h resazurin time not indicated with "*" - are they considered not significantly different?

(iii) For the Supplementary Information file, in addition to the t-values/p-values/ANOVA values, the authors should upload the raw data used to build the viability graphs.

---

## Round 0.5 · Minor Revisions

[1] In the Rebuttal letter, the authors stated 'We add the following sentence (428-429): “Also, TMX drastically reduced the viability of PBMCs obtained from healthy donors, when PCL-TMX added to the culture medium.”' But in the PDF file for Review, the phrase "PCL-TMX" is missing! The meaning of the sentence is different without this phrase since it would refer to free TMX.

[2] The authors stated that the differences in the 3 sets of data in Fig. 7b for 24, 30 and 48h resazurin time are not statistically significant, so it was not indicated with “*”. Please explain how it is possible that in these 3 sets, ~100% viability without nanofiber is not significant different from ~0% viability with PCL-TMX. I would strongly advise the authors to check their statistical calculations in the SI or consult a statistician.

[3] Similar to [2], the "*" that was originally present in Fig. 7a for t=4, 24, 30, 48 h has been removed. So are these sets also considered not to show significant differences even though there is >250% viability with PCL-P versus 100% viability without nanofiber?

---

## Round 0.6 · Minor Revisions

[1] The changes indicated in the WORD document on the issue of statistical differences were not reflected in the PDF file for review (V5). Please ensure that changes made in the WORD file are reflected in the text as well as in the Figure legends in the text and above Figures 6 and 7 of the PDF file.

[2] It is not clear what is the meaning of the change made in Lines 404-405: "those incubated with PCL-P showed a slight decrease in viability but like no treated cell"
From Figure 6, it appears "those incubated with PCL-P showed a slight decrease in viability unlike the untreated cells" since viability of untreated cells remained at ~100%. Please check and make the appropriate revision.

---

## Round 0.7 · Minor Revisions

The changes described by the authors in the Letter have been reflected in the PDF file although the WORD file peerj-56248-Manuscript_tracked_changes_3 did not reflect the changes as described in the Letter. Please ensure that the tracked changes document and the clean manuscript agree when you resubmit.

The section editor has also rightly pointed out that the final sentence of the abstract "Based on these results, the PCL-TMX nanofibers developed have potential as an alternative for local chronic TMX use for breast cancer treatment without affecting other cells or tissues." should be removed or toned down, because this work has not shown that these PCL-TMX nanofibers are harmless to other tissues, and indeed did not even test that.

In fact, Fig 7 and the discussion in the paragraph starting from Line 428 have clearly demonstrated the toxicity of PCL-TMX to PBMCs. Thus, not only should the last line of the Abstract but other claims of biocompatibility or lack of collateral effects elsewhere in the text such as Lines 395, 474 and the Conclusion, should also be removed or toned down.

Please check the manuscript carefully to ensure that no other conflicting statements are present to avoid further delays in the acceptance of this manuscript.

---

## Round 0.8 · accepted · Accept

The corrections made are satisfactory. Please note that there is a minor correction for Line 396 - "....a TMX local delivery system. Since the cytotoxic effect..." should be "...a TMX local delivery system since the cytotoxic effect..."